# Privacy-preserving Q-Learning with Functional Noise in Continuous Spaces

**Baoxiang Wang**
The Chinese University of Hong Kong
Borealis AI, Edmonton
bxwang@cse.cuhk.edu.hk

**Nidhi Hegde**
Borealis AI, Edmonton
nidhi.hegde@borealisai.com

## Abstract

We consider differentially private algorithms for reinforcement learning in continuous spaces, such that neighboring reward functions are indistinguishable. This protects the reward information from being exploited by methods such as inverse reinforcement learning. Existing studies that guarantee differential privacy are not extendable to infinite state spaces, as the noise level to ensure privacy will scale accordingly to infinity. Our aim is to protect the value function approximator, without regard to the number of states queried to the function. It is achieved by adding functional noise to the value function iteratively in the training. We show rigorous privacy guarantees by a series of analyses on the kernel of the noise space, the probabilistic bound of such noise samples, and the composition over the iterations. We gain insight into the utility analysis by proving the algorithm's approximate optimality when the state space is discrete. Experiments corroborate our theoretical findings and show improvement over existing approaches.

## 1 Introduction

Increasing interest in reinforcement learning (RL) and deep reinforcement learning has led to recent advances in a wide range of algorithms [SB18]. While a large part of the advancement has been in the space of games, the applicability of RL extends to other practical cases such as recommendation systems [ZZZ+18, LSTS15] and search engines [RJG+18, HDZ+18]. With the popularity of the RL algorithms increasing, so have concerns about their privacy. Namely, the released value (or policy) function are trained based on the reward signal and other inputs, which commonly rely on sensitive data. For example, an RL recommendation system may use the reward signals simulated by users' historical records. This historical information can thus be inferred by recursively querying the released functions. We consider differentially privacy [DMNS06, DR14], a natural and standard privacy notion, to protect such information in the RL methods.

RL methods learn by carrying out actions, receiving rewards observed for that action in a given state, and transitioning to the next states. Observation of the learned value function can reveal sensitive information: *the reward function* is a succinct description of the task. It is also connected to the users' preferences and the criteria of their decision-making; *the visited states* carry important contextual information on the users, such as age, gender, occupation, and etc.; *the transition function* includes the dynamics of the system and the impact of the actions on the environment. Among those, the reward function is the most vulnerable and valuable component, and studies have been conducted to infer this information [AN04, NR00]. In this paper, our aim is to design differentially private algorithms for RL, such that neighboring reward functions are indistinguishable.

There is a recent line of research on privacy-preserving algorithms by protecting the reward function. Balle et al. [BGP16] train the private value function using a fixed set of trajectories. However when a new state is queried this privacy guarantee will not hold. Similar results are also considered in

contextual multi-arm bandits [SS18, SS19, PFW$^{+}$18], where the context vector is analogous to the state. The gap that these works leave lead us to design a private algorithm that is not dependent on the number of states queried to the value function.

In order to achieve this under continuous space settings, we investigate the Gaussian process mechanism proposed by Hall et al. [HRW13]. The mechanism adds functional noise to the value function approximation hence the function can be evaluated at arbitrarily many states while preserving privacy. We show that our choice of the reproducing kernel Hilbert space (RKHS) embeds common neural networks, hence a nonlinear value function can also be used. We therefore adapt Q-learning [MKS$^{+}$15, WD92, Bai95] so that the value function is protected after each update, even when new states are visited.

We rigorously show differential privacy guarantees of our algorithm with a series of techniques. Notably, we derive a probabilistic bound of the sample paths thus ensuring that the RKHS norm of the noised function can be bounded. This bound is significantly better than a union bound of all noise samples. Further, we analyze the composition of the privacy costs of the mechanism. There is no known composition result of the functional mechanism, other than the general theorems that apply to any mechanism [KOV13, DRV10, BKN10]. Inspired by these theorems, we derive a privacy guarantee which is better than existing results. On the utility analysis, though there is no known performance analysis on deep reinforcement learning, we gain insights by proving the utility guarantee under the tractable discrete state space settings. Empirically, experiments corroborate our theoretical findings and show improvement over existing methods.

**Related Works.** There is a recent line of research that discusses privacy-preserving approaches on online learning and stochastic multi-armed bandit problems [SB18, Sze10]. The algorithms protect neighboring reward sequences from being distinguished, which is related to our definition of neighboring reward functions. In bandit problems, the algorithms preserve the privacy via mechanisms that add noise to the estimates of the reward distribution [TD17, TD16, MT15, TS13, KGGW15]. This line of work shares similar motivations as our work, but they do not scale to the continuous space because of the $\sqrt{N}$ or $\sqrt{N \ln N}$ factor involved where $N$ is the number of arms. Similarly, in the online learning settings, the algorithms preserve the privacy evaluated sequence of the oracle [GUK17, ALMT17, AS17, JKT12]. Their analyses are based on optimizing a fixed objective thus do not apply to our setting.

More closely related are privacy studies on contextual bandits [SS19, SS18], where there is a contextual vector that is analogous to the states in reinforcement learning. Equivalently, differentially private policy evaluation [BGP16] considers a similar setting where the value function is learned on a one-step MDP. It worth note that they also consider the privacy with respect to the state and the actions, though in this paper we will focus only on the rewards. The major challenge to extend these works is that reinforcement learning requires an iterative process of policy evaluation and policy improvement. The additional states that are queried to the value function are not guaranteed to be visited and protected by previous iterations. We propose an approach for both the evaluation and the improvement, while also extending the algorithm to nonlinear approximations like neural networks.

Differential privacy in a Markov decision process (MDP) has been discussed [Ven13] via the input perturbation technique. In the work, the reward is reformulated as a weighted sum of the utility and the privacy measure. With this formulation, it amounts to learn the MDP under this weighted reward. Essentially, input perturbation will cause relatively large utility loss and is therefore less preferred. Similarly, output perturbation can be used to preserve privacy, as shown in our analysis. It is though obvious that the necessary noise level is relatively larger and also depends on more factors than our algorithm does. Therefore, more subtle techniques will be required to improve the methods by input and output perturbation.

A general approach that can be applied to continuous spaces is the differentially private deep learning framework [ACG$^{+}$16, CBK$^{+}$19]. The method perturbs the gradient estimator in the updates of the neural network parameters to preserve privacy. In our problem, applying the method will require large noise levels. In fact, the algorithm considers neighboring inputs that at most one data point can be different, therefore benefits from a $1/B$ factor via privacy amplification [KLN$^{+}$11, BKN10] where $B$ is the batch size. This no longer holds in reinforcement learning, as all reward signals can be different for neighboring reward functions, causing the noise level to scale $B$ times back.

## 2 Preliminaries

### 2.1 Markov Decision Process and Reinforcement Learning

Markov decision process (MDP) is a framework to model decisions in an environment. We use canonical settings of the discrete-time Markov decision process. An MDP is denoted by the tuple $(\mathcal{S}, \mathcal{A}, \mathcal{T}, r, \rho_0, \gamma)$ which includes the state space $\mathcal{S}$, the action space $\mathcal{A} = \{1, \ldots, m\}$, the stochastic transition kernel $\mathcal{T} : \mathcal{S} \times \mathcal{A} \times \mathcal{S} \to \mathbb{R}^+$, the reward function $r : \mathcal{S} \times \mathcal{A} \to \mathbb{R}$, the initial state distribution $\rho_0 : \mathcal{S} \to \mathbb{R}^+$ and the discount factor $\gamma \in [0, 1)$. Denote $m$ in the above as the number of actions in the action space. The objective is to maximize the expected discounted cumulative reward. Further define the policy function $\pi : \mathcal{S}, \mathcal{A} \to \mathbb{R}^+$ and the corresponding action-state value function as

$$Q^\pi(s, a) = \mathbb{E}_\pi[\sum_{t \geq 0}^{\infty} \gamma^t r(s_t, a_t) | s_0 = s, a_0 = a, \pi].$$

When the context is clear, we omit $\pi$ and write $Q(s, a)$ instead.

We use the continuous state space setting for this paper, except in Appendix D. We investigate bounded and continuous state space $\mathcal{S} \subseteq \mathbb{R}$ and without loss of generality assume that $\mathcal{S} = [0, 1]$. The value function $Q(s, a)$ is treated as a set of $m$ functions $Q_a(s)$, where each function is defined on $[0, 1]$. The reward function is similarly written as a set of $m$ functions, each defined on $[0, 1]$. We do not impose any particular assumptions on the reward function.

Our algorithm is based on deep Q-learning [MKS$^+$15, Bai95, WD92], which solves the Bellman equation. Our differential privacy guarantee can also be generalized to other Q-learning algorithms. The objective of deep Q-learning is to minimize the Bellman error

$$\frac{1}{2}(Q(s, a) - \mathbb{E}[r + \gamma \max_{a'} Q(s', a')])^2,$$

where $s' \sim \mathcal{T}(s, a, s')$ denotes the consecutive state after executing action $a$ at state $s$. Similar to [MKS$^+$15], we use a neural network to parametrize $Q(s, a)$. We will focus on output a learned value function where the reward function $r(\cdot)$ and $r'(\cdot)$ cannot be distinguished by observing $Q(s, a)$, as long as $\|r - r'\|_\infty \leq 1$. Here without ambiguity we write $r(\cdot), r'(\cdot)$ as $r, r'$, and the infinity norm $\|f(s)\|_\infty$ is defined as $\sup_s |f(s)|$.

### 2.2 Differential Privacy

Differential privacy [DKM$^+$06, DMNS06] has developed into a strong standard for privacy guarantees in data analysis. It provides a rigorous framework for privacy guarantees under various adversarial attacks.

The definition of differential privacy is based on the notion that in order to preserve privacy, data analysis should not differ at the aggregate level whether any given user is present in the input or not. This latter condition on the presence of any user is formalized through the notion of neighboring inputs. The definition of neighboring inputs will vary according to the problem settings.

Let $d, d' \in \mathcal{D}$ be neighboring inputs.

**Definition 1.** *A randomized mechanism $\mathcal{M} : \mathcal{D} \to \mathcal{U}$ satisfies $(\epsilon, \delta)$-differential privacy if for any two neighboring inputs $d$ and $d'$ and for any subset of outputs $\mathcal{Z} \subseteq \mathcal{U}$ it holds that*

$$\mathbb{P}(\mathcal{M}(d) \in \mathcal{Z}) \leq \exp(\epsilon)\mathbb{P}(\mathcal{M}(d') \in \mathcal{Z}) + \delta.$$

An important parameter of a mechanism is the (global) sensitivity of the output.

**Definition 2.** *For all pairs $d, d' \in \mathcal{D}$ of neighboring inputs, the sensitivity of a mechanism $\mathcal{M}$ is defined as*

$$\Delta_\mathcal{M} = \sup_{d, d' \in \mathcal{D}} \|\mathcal{M}(d) - \mathcal{M}(d')\|, \tag{1}$$

where $\| \cdot \|$ is a norm function defined on $\mathcal{U}$.

**Vector-output mechanisms.** For converting vector-valued functions into a $(\epsilon, \delta)$-DP mechanism, one of the standard approaches is the Gaussian mechanism. This mechanism adds $\mathcal{N}(0, \sigma^2\mathbb{I})$ to the output $\mathcal{M}(d)$. In this case $\mathcal{U} = \mathbb{R}^n$ and $\| \cdot \|$ in (1) is the $\ell^2$-norm $\| \cdot \|_2$.

**Proposition 3** (Vector-output Gaussian mechanism; Theorem A.1 of [DR14]). *If $0 < \epsilon < 1$ and $\sigma \geq \sqrt{2 \ln(1.25/\delta)} \Delta_{\mathcal{M}}/\epsilon$, then $\mathcal{M}(d) + y$ is $(\epsilon, \delta)$-differentially private, where $y$ is drawn from $\mathcal{N}(0, \sigma^2 I)$.*

**Function-output mechanisms.** In this setting the output of the function is a function, which means the mechanism is a functional. We consider the case where $\mathcal{U}$ is an RKHS and $\|\cdot\|$ in (1) is the RKHS norm $\|\cdot\|_{\mathcal{H}}$. Hall et al. [HRW13] have shown that adding a Gaussian process noise $\mathcal{G}(0, \sigma^2 K)$ to the output $\mathcal{M}(d)$ is differentially private, when $K$ is the RKHS kernel of $\mathcal{U}$. Let $\mathcal{G}$ denote the Gaussian process distribution.

**Proposition 4** (Function-output Gaussian process mechanism [HRW13]). *If $0 < \epsilon < 1$ and $\sigma \geq \sqrt{2 \ln(1.25/\delta)} \Delta_{\mathcal{M}}/\epsilon$, then $\mathcal{M}(d) + g$ is $(\epsilon, \delta)$-differentially private, where $g$ is drawn from $\mathcal{G}(0, \sigma^2 K)$ and $\mathcal{U}$ is an RKHS with kernel function $K$.*

Note that in [HRW13] the stated condition was $\sigma \geq \sqrt{2 \ln(2/\delta)} \Delta_{\mathcal{M}}/\epsilon$. The improvement from constant 2 to 1.25 is natural but for the completeness we include a proof in Appendix B.

## 3 Differentially Private Q-Learning

### 3.1 Our Algorithm

We present our algorithm for privacy-preserving Q-learning under the setting of continuous state space in Algorithm 1. The algorithm is based on deep Q-learning proposed by Mnih et al. [MKS+15]. We achieve privacy by perturbing the learned value function at each iteration, by adding a Gaussian process noise. The noise-adding is described by line 19-20 of the algorithm, where $\hat{g}$ is the noise. This noise is a discrete estimate of the continuous sample path, evaluated at the states $s_t$ visited in the trajectories. Intuitively, when $(s, z)$ is an element of the list $\hat{g}$, it implies $g(s) = z$ for the sample path $g$. Line 14-18 describes the necessary maintenance of $\hat{g}$ to simulate the Gaussian process. Line 7-9 samples a new Gaussian process sample path for every $J$ iterations, which controls the balance between the approximation factor of privacy and the utility. The other steps are similar to [MKS+15].

---

**Algorithm 1** Differentially Private Q-Learning with Functional Noise

1: **Input:** the environment and the reward function $r(\cdot)$
2: **Parameters:** target privacy $(\epsilon, \delta)$, time horizon $T$, batch size $B$, action space size $m$, learning rate $\alpha$, reset factor $J$
3: **Output:** trained value function $Q_\theta(s, a)$
4: **Initialization:** $s_0 \in [0, 1]$ uniformly, $Q_\theta(s, a)$ for each $a \in [m]$, linked list $\hat{g}_k[B][2] = \{\}$
5: Compute noise level $\sigma = \sqrt{2(T/B) \ln(e + \epsilon/\delta)} C(\alpha, k, L, B)/\epsilon$;
6: **for** $j$ **in** $[T/B]$ **do**
7:    **if** $j \equiv 0 \mod T/JB$ **then**
8:       $\hat{g}_k[B][2] \leftarrow \{\}$;
9:    **end if**
10:   **for** $b$ **in** $[B]$ **do**
11:      $t \leftarrow jT/B + b$;
12:      Execute $a_t = \arg\max_a Q_\theta(s_t, a) + \hat{g}_a(s_t)$;
13:      Receive $r_t$ and $s_{t+1}$, $s \leftarrow s_{t+1}$;
14:      **for** $a \in [m]$ **do**
15:        Insert $s$ to $\hat{g}_a[:][1]$ such that the list remains monotonically increasing;
16:        Sample $z_{at} \sim \mathcal{N}(\mu_{at}, \sigma d_{at})$, according to Equation (2), Appendix A;
17:        Update the list $\hat{g}_a(s) \leftarrow z_{at}$;
18:      **end for**
19:      $y_t \leftarrow r_t + \gamma \max_a Q_\theta(s_{t+1}, a) + \hat{g}_a(s_{t+1})$;
20:      $l_t \leftarrow \frac{1}{2}(Q_\theta(s_t, a_t) + \hat{g}_a(s_t) - y_t)^2$;
21:   **end for**
22:   Run one step SGD $\theta \leftarrow \theta + \alpha \frac{1}{B} \nabla_\theta \sum_{t=jB}^{(j+1)B} l_t$;
23: **end for**
24: Return the trained $Q_\theta(s, a)$ function;

---

**Insight into the algorithm design.** To satisfy differential privacy guarantees, we require two reward functions $r$ and $r'$ to be indistinguishable upon observation of the learned functions, as long as $\|r - r'\|_\infty \leq 1$. The major difficulty is that the reward signal $r(s, a)$ can appear at any $s$, and all the reward signals can be different under $r$ and $r'$. Therefore, we will need a stronger mechanism of privacy that does not rely on the finite setting where at most one data point in a (finite) dataset is different, like in [ACG+16] and [BGP16]. This is also the major challenge in extending [BGP16] from policy evaluation to policy improvement. The natural approach to address the challenge is to treat a function as one "data point", which leads to our utilization of the techniques studied by Hall et al. [HRW13].

## 3.2   Privacy, Efficiency, and Utility of the Algorithm

**Privacy analysis.** There are three main components in the privacy analysis. First, we have to define the RKHS to invoke the Gaussian process mechanism in Proposition 4. This RKHS should also include the value function approximation we used in the algorithm, namely, neural networks. Second, we give a privacy guarantee of composing the mechanism for $T/B$ iterations. There is not a known composition result of such a functional mechanism, other than the general theorems that apply to any mechanism [KOV13, DRV10, BKN10]. But we derive such a privacy guarantee which is better than existing results. Third, as the sample path is evaluated on multiple different states, the updated value function can be unbounded, which subsequently induces the RKHS norm to be unbounded. This is addressed by showing a probabilistic uniform bound of the sample path over the state space.

Our privacy guarantee is shown in the following theorem.

**Theorem 5.** *The Q-learning algorithm in Algorithm 1 is $(\epsilon, \delta + J \exp(-(2k - 8.68\sqrt{\beta}\sigma)^2/2))$-DP with respect to two neighboring reward functions $\|r - r'\|_\infty \leq 1$, provided that $2k > 8.68\sqrt{\beta}\sigma$, and*

$$\sigma \geq \sqrt{2(T/B)\ln(e + \epsilon/\delta)}C(\alpha, k, L, B)/\epsilon,$$

*where $C(\alpha, k, L, B) = ((4\alpha(k+1)/B)^2 + 4\alpha(k+1)/B)L^2$, $\beta = (4\alpha(k+1)/B)^{-1}$, $L$ is the Lipschitz constant of the value function approximation, $B$ is the batch size, $T$ is the number of iterations, and $\alpha$ is the learning rate.*

Theorem 5 provides a rigorous guarantee on the privacy of the reward function. We now present three statements to address the challenges mentioned above and support the theorem.

Lemma 6 and its corollary, informally stated below and formally stated in Appendix C, describe the RKHS that is necessary to both embedded the function approximators we use and invoke the mechanism in Proposition 4.

**Lemma 6** (Informal statement). *The Sobolev space $H^1$ with order $1$ and the $\ell^2$-norm is defined as*

$$H^1 = \{f \in C[0,1] : \partial f(x) \text{ exists}; \int_0^1 (\partial f(x))^2 dx < \infty\},$$

*where $\partial f(x)$ denotes weak derivatives and the RKHS kernel is $K(x, y) = \exp(-\beta|x - y|)$.*

Immediately following Lemma 6, we show that the common neural networks are in the Sobolev space. That includes neural networks with nonlinear activation layers such as a ReLU function, a sigmoid function, or the tanh function. The proof of the following corollary is also in Appendix C.

**Corollary 7.** *Let $\hat{f}_W(x)$ denote the neural network with finitely many finite parameters $W$. For $\hat{f}_W(x)$ with finitely many layers, if the gradient of the activation function is bounded, then $\hat{f}_W(x) \in H^1$.*

By the corollary $\hat{f}_W(x)$ is Lipschitz continuous. Denote $L$ as the Lipschitz constant which only depends on the network architecture. It follows from Lemma 6 immediately that, in the algorithm for any $Q(s, a)$ and $Q'(s, a)$, $\|Q(\cdot, a) - Q'(\cdot, a)\|_{\mathcal{H}}^2 \leq 2r_0^2(1 + \beta/2)/(1 - \gamma)^2 + L^2/\beta$, for each $a$, where it assumes bounded reward $|r(s, a)| \leq r_0$. This will lead to an alternative privacy guarantee, but less preferred than in Theorem 5 due to the $1/(1 - \gamma)^2$ and the $r_0$ factor.

Line 19 and 20 use $\hat{g}$, which is the list of Gaussian random variables evaluated at the Gaussian process sample paths. Using a union tail bound we can derive a probabilistic bound of these variables, but it will cause the approximation factor to be $\delta + \mathcal{O}(1 - (1 - \exp(2k - \sqrt{\beta}\sigma))^T)$, which is unrealistically

large. We show in the lemma below that with high probability the entire sample path is uniformly bounded over any state $s_t$. We can then calibrate the $\delta$ to cover the exponentially small tail bound $\mathcal{O}(\exp(-u^2))$ of the noise. The proof is in Appendix A.

**Lemma 8.** *Let $\mathbb{P}$ the probability measure over $H^1$ of the sample path $f$ generated by $\mathcal{G}(0, \sigma^2 K)$. Then almost surely $\max_{x \in [0,1]} f(x)$ exists, and for any $u > 0$*

$$\mathbb{P}(\max_{x \in [0,1]} f(x) \geq 8.68\sqrt{\beta}\sigma + u) \leq \exp(-u^2/2).$$

*Proof of Theorem 5.* Let $Q$ and $Q'$ denote the learned value function of the algorithm given $r$ and $r'$, respectively, where $\|r - r'\|_\infty \leq 1$. To make $Q$ and $Q'$ indistinguishable, we inspect the update step in line 21. Let $Q_0$ denote the value function after and before the update, we have

$$\|Q - Q_0\|_\infty \leq \alpha L(2 + \hat{g}_a(s_{t+1}) - \hat{g}_a(s_t))/B.$$

As per Lemma 8, with probability at least $1 - \exp(-(2k - 8.68\sqrt{\beta}\sigma)^2/2)$, we have $|Q - Q_0| \leq 2\alpha L(k+1)/B$. By the triangle inequality, for any $\|r - r'\|_\infty \leq 1$, the corresponding $Q$ and $Q'$ satisfies $\|Q - Q'\|_\infty \leq 4\alpha L(k+1)/B$, given that $Q_0$ is fixed by the previous step. Let $f = Q - Q'$, we have

$$\|f\|_{\mathcal{H}}^2 \leq (1 + \beta/2)(4\alpha L(k+1)/B)^2 + L^2/2\beta$$

by the formal statement Lemma 6, Appendix C. We choose $1/\beta = 4\alpha(k+1)/B$ and have $\|f\|_{\mathcal{H}}^2 \leq ((4\alpha(k+1)/B)^2 + 4\alpha(k+1)/B)L^2$. Now by Proposition 4, adding $g \sim \mathcal{G}(0, \sigma^2 K)$ to $Q$ will make the update step $(\epsilon', \delta' + \exp(-(2k - 8.68\sqrt{\beta}\sigma)^2/2))$-differentially private, given that $\sigma \geq \sqrt{2 \ln(1.25/\delta')}\|f\|_{\mathcal{H}}/\epsilon'$, where $K(x, y) = \exp(-4\alpha L(k+1)|x - y|/B)$ is our choice of the kernel function. Thus each iteration of update has a privacy guarantee.

It amounts to analyze the composition of $T/B$ many iterations. It is shown by the composition theorem [KOV13, Mir17] that any $\sigma \geq \sqrt{2(T/B) \ln(1.25/\delta) \ln(e + \epsilon/\delta)}\|f\|_{\mathcal{H}}/\epsilon$ is sufficient. This is the best known bound, but we continue to derive the specific bound for our algorithm. Let $z$ (a function, either $Q$ or $Q'$) be the output of a single update of the algorithm. Denote $v = 4\alpha(k+1)/B$ and $T' = T/B$ for simplicity. By Lemma 11, Appendix A, we have

$$\mathbb{E}_0[(\mathbb{P}_1(z \in S)/\mathbb{P}_0(z \in S))^\lambda] \leq \exp(\frac{(\lambda^2 + \lambda)(v^2 + v)L^2}{2\sigma^2}),$$

where $\mathbb{P}_0$ and $\mathbb{P}_1$ are the probability distribution of $z$ given $r$ and $r'$, respectively. This moment generating function will scale exponentially if multiple independent instances of $z$ are drawn. Namely, let $\boldsymbol{z}$ be the vector of $T'$ many independent $z$, and $\mathbb{P}_0^{T'}$ and $\mathbb{P}_1^{T'}$ be its probability distribution under $r$ and $r'$. We have for $\lambda > 0$, $\mathbb{E}_0[(\mathbb{P}_1^{T'}(\boldsymbol{z} \in S)/\mathbb{P}_0^{T'}(\boldsymbol{z} \in S))^\lambda] \leq \exp(\frac{(\lambda^2+\lambda)(v^2+v)L^2}{2\sigma^2}T')$. Thus,

$$\exp(\lambda(\ln(\mathbb{P}_1^{T'}(\boldsymbol{z})/\mathbb{P}_0^{T'}(\boldsymbol{z})) - \epsilon) = \exp(\frac{T'\|f\|_{\mathcal{H}}^2}{2\sigma^2}(\lambda + \frac{1}{2}(1 - \frac{2\epsilon\sigma^2}{T'\|f\|_{\mathcal{H}}^2}))^2 - \frac{1}{4}(1 - \frac{2\epsilon\sigma^2}{T'\|f\|_{\mathcal{H}}^2})^2).$$

Since the argument holds for any $\lambda > 0$, let $\lambda = -\frac{1}{2}(1 - \frac{2\epsilon\sigma^2}{T'\|f\|_{\mathcal{H}}^2}) > 0$, then

$$\begin{aligned}
\mathbb{P}_1^{T'}(\boldsymbol{z}) - \exp(\epsilon)\mathbb{P}_0^{T'}(\boldsymbol{z}) &\leq \mathbb{E}_0[\exp(\lambda(\ln(\mathbb{P}_1^{T'}(\boldsymbol{z})/\mathbb{P}_0^{T'}(\boldsymbol{z})) - \epsilon) + \lambda\ln\lambda - (\lambda+1)\ln(\lambda+1))] \\
&= \exp(-\frac{T'\|f\|_{\mathcal{H}}^2}{2\sigma^2}(1 - \frac{2\sigma^2}{T'\|f\|_{\mathcal{H}}^2}\epsilon)^2 + \lambda\ln\lambda - (\lambda+1)\ln(\lambda+1)) \\
&\leq \exp(-\frac{T'\|f\|_{\mathcal{H}}^2}{2\sigma^2}(1 - \frac{2\sigma^2}{T'\|f\|_{\mathcal{H}}^2}\epsilon)^2)(\lambda+1) \\
&= \exp(-\frac{\sigma^2}{2T'\|f\|_{\mathcal{H}}^2}(\epsilon - \frac{T'\|f\|_{\mathcal{H}}^2}{2\sigma^2})^2)\frac{1}{1 + \frac{\sigma^2}{T'\Delta^2}(\epsilon - \frac{T'\|f\|_{\mathcal{H}}^2}{2\sigma^2})}.
\end{aligned}$$

We desire to find $\epsilon$ and $\delta$ so that this difference $\mathbb{P}_1^{T'}(\boldsymbol{z}) - \exp(\epsilon)\mathbb{P}_0^{T'}(\boldsymbol{z})$ is less than $\delta$. We use similar techniques as is in the proof Theorem 4.3 of [KOV13]. We choose

$$\epsilon = \frac{T'\|f\|_{\mathcal{H}}^2}{2\sigma^2} + \sqrt{\frac{2T'\|f\|_{\mathcal{H}}^2 w}{\sigma^2}},$$

where $w = \ln(e + \sqrt{T'\|f\|_{\mathcal{H}}^2/\sigma^2}/\delta)$. Thus the first term is $e^{-w}$ and the second term is $\frac{1}{1+\sqrt{\frac{2\sigma^2 w}{T'\|f\|_{\mathcal{H}}^2}}}$.

This ensures that $e^{-w} \leq \delta/\sqrt{T'\|f\|_{\mathcal{H}}^2/\sigma^2}$ and $\frac{1}{1+\sqrt{\frac{2\sigma^2 w}{T'\|f\|_{\mathcal{H}}^2}}} \leq \frac{1}{1+\sqrt{\frac{2\sigma^2}{T'\|f\|_{\mathcal{H}}^2}}}$, thereby guaranteeing

that $\mathbb{P}_1^{T'}(\boldsymbol{z}) - \exp(\epsilon)\mathbb{P}_0^{T'}(\boldsymbol{z}) \leq \delta$ for differential privacy. We solve $\epsilon = \frac{T'\|f\|_{\mathcal{H}}^2}{2\sigma^2} + \sqrt{\frac{2T'\|f\|_{\mathcal{H}}^2 w}{\sigma^2}}$ and find the sufficient condition that

$$\sigma = \sqrt{(2T'\|f\|_{\mathcal{H}}^2/\epsilon^2)\ln(e + \epsilon/\delta)}$$
$$\leq \sqrt{2(T/B)((4\alpha(k+1)/B)^2 + 4\alpha(k+1)/B)L^2 \ln(e + \epsilon/\delta)}/\epsilon,$$

as desired. When this sufficient condition is satisfied, the approximation factor will be no larger than $\delta$ plus $J$ times the uniform bound derived above by Lemma 8. Namely, it achieves $(\epsilon, \delta + J\exp(-(2k - 8.68\sqrt{\beta}\sigma)^2/2))$-DP. $\qquad\square$

**Time complexity.** We show that the noise adding mechanism in our algorithm is efficient. In fact, the most complex step introduced by the noise-adding is the insertion in line 15. This can be negligible compared with the steps such as computing gradients and executing actions. A complete proof of the below proposition is given in Appendix A.

**Proposition 9.** *The noised value function (during either training or released) in Algorithm 1 can respond to $N_q$ queries in $\mathcal{O}(N_q \ln(N_q))$ time.*

**Utility analysis.** To the best of our knowledge, there has not been a study to rigorously analyze the utility of deep reinforcement learning. In fact, in the continuous state space setting, the solution of the Bellman equation is not unique in general. Hence, it is unlikely for Q-learning to achieve a guaranteed performance, even if it converges. However, we gain insights by analyzing the algorithm's learning error in the discrete state space setting. The learning error is defined by the discrepancy between the learned state value function and the ground truth of the optimal state value function. We consider the worst case $J = 1$ for utility (which is the best case for $(\epsilon, \delta + \exp(-(2k - 8.68\sqrt{\beta}\sigma)^2/2))$-differential privacy) where the noise is the most correlated through the iterations. We show the upper bound of the utility loss, which has a limit of zero as the number of states approaches infinity. The proof involves the linear program formulation of MDP, which is given in Appendix D.

**Proposition 10.** *Let $v'$ and $v^*$ be the value function learned by our algorithm and the optimal value function, respectively. In the case $J = 1$, $|S| = n < \infty$, and $\gamma < 1$, the utility loss of the algorithm satisfies*

$$\mathbb{E}[\frac{1}{n}\|v' - v^*\|_1] \leq \frac{2\sqrt{2}\sigma}{\sqrt{n\pi}(1-\gamma)}.$$

## 3.3 Discussion

**Extending to other RL algorithms.** Our algorithm can be extended to the case where it learns both a policy function and a value function coordinately, like the actor-critic method [MBM+16]. If in the updates of the policy function, only the $Q$ function is used in the policy gradient estimation, for example $\nabla_{\theta_\pi} \ln\pi(a|s)Q(s,a)$, then the algorithm has the same privacy guarantee. Also, any post-processing of the private $Q$ function will not break the privacy guarantee. This also includes experience replay and $\epsilon$-greedy policies [MKS+15].

However, consider the case where the reward is directly accessed in the policy gradient estimation, for example $\nabla_{\theta_\pi} \ln\pi(a|s)A(s,a)$ in [DWS12, SML+15] where $A(s,a) = \sum_{t=0}^{T}(\lambda\gamma)^t(r_t + v(s_{t+1}) - v(s_t))$. In this setting, the privacy guarantee no longer holds. To extend the privacy guarantee to this case, one should add noise to the policy function as well.

**Extending to high-dimensional tasks.** We assumed in our analysis $\mathcal{S} = [0, 1]$ for simplicity. The setting extends to any bounded $\mathcal{S} \subseteq \mathbb{R}$ by scaling the interval. Our approach can also be extended to high-dimensional spaces by choosing a high-dimensional RKHS and kernel. For example, the kernel function $\exp(-\beta|x - y|)$ where $x$ and $y$ now belongs to $R^n$ and $|\cdot|$ is the Manhattan distance. It is also possible to use other RKHS and kernels for the Gaussian process noise, such as the space of band-limited functions. Other than the re-calibration of the noise level to the new kernel, the privacy guarantee in the theorem holds in general for the respective definition of $\|\cdot\|_{\mathcal{H}}$. We note that the time

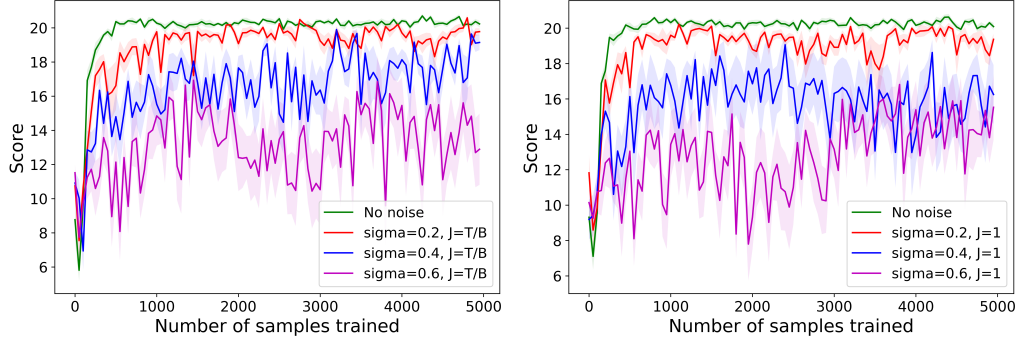

Figure 1: Empirical results of Algorithm 1 on different noise levels. The y-axis denotes the return. The x-axis is the number of samples the agent has trained on. Each episode has 50 samples. The shadow denotes 1-std. The learning curves are averaged over 10 random seeds. The curves are generated without smoothing.

complexity derived in Proposition 9 does not extend to other kernel functions, which requires the algorithm to take $\mathcal{O}(N_q^2)$ in the noise generating process.

## 4 Experiments

We present empirical results to corroborate our theoretical guarantees and to demonstrate the performance of the proposed algorithm on a small example. The exact MDP we use is described in Appendix E.1. The implementation is attached along with the manuscript submission.

We first plot the learning curve with a variety of noise levels and $J$ values in Figure 1. With the increase of the noise level, the algorithm requires more samples to achieve the same return than the non-private version. This demonstrates the empirical privacy-utility tradeoff. We observe that with the noise being reset every round ($J = T/B$), the algorithm is likely to converge with limited sub-optimality as desired, especially when $\sigma < 0.4$. Therefore, as $J \exp(-(2k - 8.68\sqrt{\beta}\sigma)^2/2)$ is exponentially small, we suggest using $J = T/B$ in practice to achieve a better utility.

The algorithm is then compared with a variety of baseline methods where they target the same $(\epsilon, \delta)$ privacy guarantee, as shown in Figure 2(a) and 2(b). The policy evaluation method proposed by Balle, Gomrokchi and Precup [BGP16] is not differentially private under our context (while it is $(\epsilon, \delta)$-DP with respect to the reward sequences). We compare with it to illustrative the utility, where it is observed that their approach shares similar performance with ours. Note that studies on contextual bandits by Sajed and Sheffet [SS19] and by Shariff and Sheffet [SS18] consider an equivalently one-step MDP as [BGP16] and thus will yield the same method. We also compared our approach with the input perturbation method proposed by Venkitasubramaniam [Ven13] and the differentially private deep learning framework by Abadi et al. [ACG$^+$16]. Both the approaches are differentially private under our setting, while our algorithm significantly outperforms them. Especially, on the higher privacy regime $\epsilon = 0.45$, both the baseline methods do not improve over the training due the the large noise level needed. The baseline implementations and the exact calculation of the parameters are detailed in Appendix E.2 and E.3, respectively.

## 5 Conclusion

We have developed a rigorous and efficient algorithm for differentially private Q-learning in continuous state space settings. Releasing and querying the algorithm's output value function will not distinguish two neighboring reward functions. To achieve this, our method applies functional noise taken from sample paths of a Gaussian process calibrated appropriately according to sensitivity calculated under the RKHS measure. Theoretically, we show the privacy guarantee and insights into the utility analysis. Empirically, experiments corroborate our theoretical findings and show improvement over existing methods. Our approach is general enough to be extended to other domains beyond reinforcement learning.

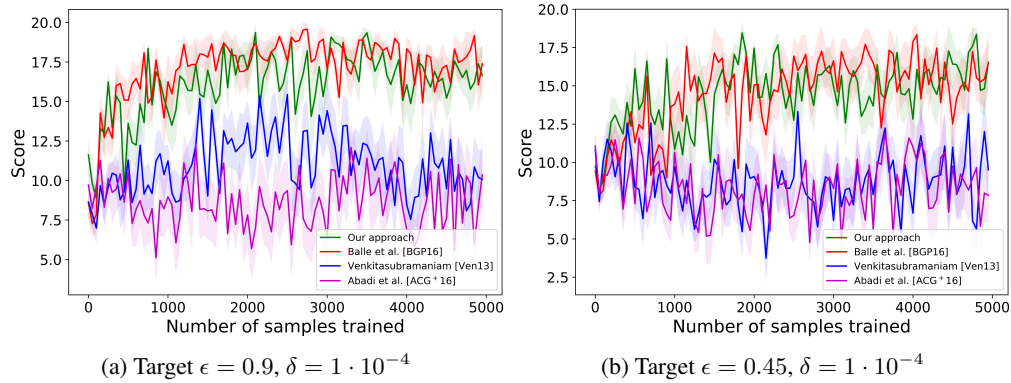

(a) Target $\epsilon = 0.9$, $\delta = 1 \cdot 10^{-4}$        (b) Target $\epsilon = 0.45$, $\delta = 1 \cdot 10^{-4}$

Figure 2: Empirical comparisons with other methods. Same configurations as Figure 1.

## Acknowledgement

We would like to thank Ruitong Huang, who provides helpful insight on the composition analysis and the algorithm design, and Kry Yik Chau Lui, who points out the idea to extend our approach to high-dimensional Sobolev spaces.

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
