[Supplementary Material]

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

# A    Proof of Lemma 8, Lemma 11 and Proposition 9

In the proofs, we will refer to some properties of the Gaussian process and its sample paths. We put those properties in Claim 12, at the end of this section. The claim and the notation used in the claim are reused in the proof of Lemma 8, Lemma 11, and Proposition 9. For simplicity, when the result can be verified immediately, such as calculating an inverse matrix, we will omit the steps.

**Notations for this section.**    We investigate a dyadic rational. For a sample path $f$ and $n \geq 1$, define $f_{n0} = \{f(x_0), f(x_2), \ldots, f(x_{2n})\}$ and $f_{n1} = \{f(x_1), f(x_3), \ldots f(x_{2n-1})\}$, where $x_i = i/2n$, $i = 0, \ldots, 2n$. For a deterministic function $g$ defined on $[0, 1]$, define $g_{n0}$ and $g_{n1}$ similarly as $g_{n0} = (\lim_{x \to x_0} g(x), \lim_{x \to x_2} g(x), \ldots, \lim_{x \to x_{2n}} g(x))^T$ and $g_{n1} = (\lim_{x \to x_1} g(x), \lim_{x \to x_3} g(x), \ldots \lim_{x \to x_{2n-1}} g(x))^T$. Also define $\beta_n = \beta/2n$. Our goal is to investigate the desired properties when $\lim_{n \to \infty}$.

By the definition of the Gaussian process,

$$\begin{pmatrix} f_{n1} \\ f_{n0} \end{pmatrix} \sim \mathcal{N}\left(0, \sigma^2 \begin{bmatrix} K_{11} & K_{10} \\ K_{10}^T & K_{00} \end{bmatrix}\right),$$

where $K_{11}$, $K_{10}$, and $K_{00}$ are depending on $n$. If $f \sim \mathcal{G}(0, \sigma^2 K)$, the conditional distribution $f_{n1}|f_{n0} \sim \mathcal{N}(K_{10}K_{00}^{-1}f_{n0}, \sigma^2(K_{11} - K_{10}K_{00}^{-1}K_{10}^T))$. If $f \sim \mathcal{G}(g, \sigma^2 K)$, $f_{n1}|f_{n0} \sim \mathcal{N}(g_{n1} + K_{10}K_{00}^{-1}(f_{n0} - g_{n0}), \sigma^2(K_{11} - K_{10}K_{00}^{-1}K_{10}^T))$.

We restate Lemma 8 and prove it. Recall that the lemma proves a tail bound for the maximum of a GP sample path.

**Lemma 8.** *Let $\mathbb{P}$ the probability measure over $H^1$ of the sample path $f$ generated by $\mathcal{G}(0, \sigma^2 K)$. Then almost surely $\max_{x \in [0,1]} f(x)$ exists, and for any $u > 0$*

$$\mathbb{P}(\max_{x \in [0,1]} f(x) \geq 8.68\sqrt{\beta}\sigma + u) \leq \exp(-u^2/2).$$

*Proof.* We start with the base case that $f_{10} = \{f(0), f(1)\}$. $\mathbb{E}[\max f_{10}] = \frac{1}{2}\mathbb{E}[|f(0) - f(1)|] = \sqrt{(1 - \exp(-\beta))/\pi}\sigma \leq \sqrt{\beta/\pi}\sigma$, where the second equation is due to $f(0) - f(1) \sim \mathcal{N}(0, 2(1 - \exp(-\beta))\sigma^2)$.

We desire to bound the expectation $\mathbb{E}[\max f_{n0}]$ for all $n > 1$. To complete this, we first prove a general result on the expectation of the maximum of the Gaussian variables. Let $z_1, \ldots, z_n \sim \mathcal{N}(0, \sigma_z^2)$ be $n$ independent Gaussian random variables, by the Chernoff bound we have

$$\exp(t\mathbb{E}[\max_i z_i]) \leq \mathbb{E}[\exp(t \max_i z_i)] = \mathbb{E}[\max_i \exp(tz_i)] \leq n \exp(t^2\sigma_z^2/2).$$

By choosing $t = \sqrt{2 \ln n}/\sigma_z$, we conclude that $\mathbb{E}[\max_i z_i] \leq \sqrt{2 \ln n}\sigma_z$. Meanwhile, it is obvious that $\mathrm{Var}[\max_i z_i] \leq \sigma_z$.

Denote $\mu_n = \mathbb{E}[\max f_{2^n0}]$. As $f_{2^n0} \subset f_{2^{n+1}0}$, the series $\mu_n$ is non-decreasing. We derive an upperbound of $\mu_{n+1} - \mu_n$. Let $x_i = i/2^{n+1}$ and $\xi_{i,n} = f(x_{2i-1}) - \frac{\exp(-\beta_{2^n})}{1+\exp(-2\beta_{2^n})}(f(x_{2i-2})+f(x_{2i}))$, we have that $\xi_{i,n} \sim \mathcal{N}(0, \sigma^2 \frac{1-\exp(-2\beta_{2^n})}{1+\exp(-2\beta_{2^n})})$. Further, $\xi_{i,n}$ and $\xi_{j,n}$ are independent. It is true as $K_{10}K_{00}^{-1}$ is nonzero only at its two diagonals, which indicates that $f(x_{2j-1})$ is not depending on other point if $f(x_{2j-2}) + f(x_{2j})$ is given. Thus, we have $\xi_{i,n}$ i.i.d. for $i = 1, \ldots, 2^n$.

As we shown before the upper bound $\mathbb{E}[\max_i z_i] \leq 2\sqrt{\ln n}\sigma_z$, the expectation is monotonically increasing on the variance $\sigma_z$. In general, we have $(1 - \exp(cx))/(1 + \exp(cx)) < x$ for all positive $x$, if and only if $c \leq 2$. Thus $\frac{1-\exp(-2\beta_{2^n})}{1+\exp(-2\beta_{2^n})} < \beta_{2^n}$, and consequently we have $\mathbb{E}[\max_i \xi_{i,n}] \leq \mathbb{E}[\max_i \xi_{i,n}']$, for $\xi_{i,n}' \sim \mathcal{N}(0, \sigma^2\beta_{2^n})$.

By the inequality $\exp(-x)/(1 + \exp(-2x)) < \frac{1}{2}$ for $x > 0$, we relax that $\frac{\exp(-\beta_n)}{1+\exp(-2\beta_n)} < \frac{1}{2}$. Thus, $f(x_{2i-1}) \leq \xi_{i,n} + \frac{1}{2}(f(x_{2i-2}) + f(x_{2i}))$. Taking maximum and expectation on both sides, we have

$$\mathbb{E}[\max_{1 \leq i \leq 2^n} f(x_{2i-1})] \leq \mathbb{E}[\max_{1 \leq i \leq 2^n} \xi_{i,n} + \frac{1}{2}(f(x_{2i-2}) + f(x_{2i}))]$$

$$\leq \mathbb{E}[\max_{1\leq i\leq 2^n} \xi'_{i,n}] + \frac{1}{2}\mathbb{E}[\max_{1\leq i\leq 2^n} f(x_{2i-2})] + \frac{1}{2}\mathbb{E}[\max_{1\leq i\leq 2^n} f(x_{2i})]$$

$$\leq \sqrt{2\ln 2^n}\sqrt{\sigma^2\beta_{2^n}} + \frac{1}{2}\mu_n + \frac{1}{2}\mu_n$$

$$= \mu_n + \sqrt{n\beta/2^n}\sigma.$$

Thus, $\mathbb{E}[\max f_{2^n 1} - \max f_{2^n 0}] \leq \sqrt{n\beta/2^n}\sigma$. Meanwhile, we have that $\mathrm{Var}[\max f_{2^n 1} - \max f_{2^n 0}] \leq \mathrm{Var}[\max f_{2^n 1}] + \mathrm{Var}[\max f_{2^n 0}] \leq 2(\beta/2^n)\sigma^2$. Let $z$ be a random variable subject to $\mathbb{E}[z] = \sqrt{n\beta/2^n}\sigma$, $\mathrm{Var}[z] = 2(\beta/2^n)\sigma^2$, then there exists a $z$ such that $\mathbb{E}[\max(0, \max f_{2^n 1} - \max f_{2^n 0})] \leq \mathbb{E}[\max(0, z)]$. We will bound this value. Denote $c = \sqrt{\beta/2^n}\sigma$ for simplicity. Then we have

$$\exp(\frac{1}{c}\mathbb{E}[\max(z,0)]) \leq \mathbb{E}[\exp(\frac{1}{c}\max(z,0))]$$

$$\leq \mathbb{E}[\max(\exp(\frac{z}{c}), \exp(0))]$$

$$\leq \mathbb{E}[\exp(\frac{z}{c} + 1)]$$

$$\leq \exp(\sqrt{n} + 1) + 1.$$

Consequently,

$$\mathbb{E}[\max(z,0)] \leq (\sqrt{n} + 1 + \frac{1}{\exp(\sqrt{n}+1)})\sqrt{\beta/2^n}\sigma.$$

Hence we have

$$\mu_{n+1} - \mu_n = \mathbb{E}[\max(0, \max f_{2^n 1} - \max f_{2^n 0})] \leq (\sqrt{n} + 1 + \frac{1}{\exp(\sqrt{n}+1)})\sqrt{\beta/2^n}\sigma.$$

By induction, we have $\mu_n \leq \sqrt{\beta/\pi}\sigma + \sum_{i=0}^{\infty}(\sqrt{i} + 1 + \frac{1}{\exp(\sqrt{i}+1)})\sqrt{\beta/2^i}\sigma < 8.68\sqrt{\beta}\sigma$, for any integer $n$. Since the dyadic rational $\cup_{i=0}^{\infty} f_{2^i 0}$ is dense and compact on $[0,1]$ and $f$ is continuous with probability one, $\mathbb{E}[\max f]$ shares the same upper bound of $\mu_n$ almost surely. It is shown in Theorem 3 of [Lal13], that if the expectation $\mathbb{E}[\max f]$ is bounded then $\max f$ is sub-Gaussian. The lemma follows. $\qquad\square$

The following Lemma 11 shows an upper bound of the moment generating function of the Gaussian process with the kernel introduced in Lemma 6. The lemma will be used to compose the Gaussian process mechanisms in Theorem 5.

**Lemma 11.** *Let $g \in H^1$ be continuous almost everywhere. Denote $\mathbb{P}_0(f)$ and $\mathbb{P}_1(f)$ be the probability measure over $H^1$ of the sample path generated by $\mathcal{G}(0, \sigma^2 K)$ and $\mathcal{G}(g, \sigma^2 K)$, respectively. The sample path $f \sim \mathbb{P}_0$ satisfies, for any $\lambda > 0$ and any set $S$ of sample paths,*

$$\mathbb{E}_0[(\mathbb{P}_1(f \in S)/\mathbb{P}_0(f \in S))^{\lambda}] \leq \exp((\lambda^2 + \lambda)\|g\|_{\mathcal{H}}^2/2\sigma^2).$$

*Proof.* By the conditional distribution $f_{n1}|f_{n0}$ in Claim 12, we have the ratio of the probability density

$$\ln\frac{\mathbb{P}_1(f_{n1}|f_{n0})}{\mathbb{P}_0(f_{n1}|f_{n0})} = -\frac{(g_{n1} - K_{10}K_{00}^{-1}g_{n0})^T(K_{11} - K_{10}K_{00}^{-1}K_{10}^T)^{-1}(g_{n1} - K_{10}K_{00}^{-1}g_{n0})}{2\sigma^2}$$

$$+ \frac{2(g_{n1} - K_{10}K_{00}^{-1}g_{n0})^T(K_{11} - K_{10}K_{00}^{-1}K_{10}^T)^{-1}(f_{n1} - K_{10}K_{00}^{-1}f_{n0})}{2\sigma^2}$$

$$= -\frac{1 + \exp(-2\beta_n)}{2\sigma^2(1 - \exp(-2\beta_n))}\sum_{i=1}^n (g(x_{2i-1}) - \frac{\exp(-\beta_n)}{1 + \exp(-2\beta_n)}(g(x_{2i-2}) + g(x_{2i})))^2$$

$$+ \frac{1 + \exp(-2\beta_n)}{\sigma^2(1 - \exp(-2\beta_n))}\sum_{i=1}^n (g(x_{2i-1}) - \frac{\exp(-\beta_n)}{1 + \exp(-2\beta_n)}(g(x_{2i-2}) + g(x_{2i})))$$

$$\times (f(x_{2i-1}) - \frac{\exp(-\beta_n)}{1 + \exp(-2\beta_n)}(f(x_{2i-2}) + f(x_{2i}))).$$

Note that in the equation above $f(x_{2i-1}) - \frac{\exp(-\beta_n)}{1+\exp(-2\beta_n)}(f(x_{2i-2}) + f(x_{2i}))$ and $f(x_{2j-1}) - \frac{\exp(-\beta_n)}{1+\exp(-2\beta_n)}(f(x_{2j-2}) + f(x_{2j}))$ are i.i.d. with the distribution $\mathcal{N}(0, \sigma^2 \frac{1-\exp(-2\beta_n)}{1+\exp(-2\beta_n)})$. Hence, We have

$$\mathbb{E}_0[\exp(\lambda \ln \frac{\mathbb{P}_1(f_{n1}|f_{n0})}{\mathbb{P}_0(f_{n1}|f_{n0})})]$$

$$= \exp(-\lambda \frac{1+\exp(-2\beta_n)}{2\sigma^2(1-\exp(-2\beta_n))} \sum_{i=1}^{n}(g(x_{2i-1}) - \frac{\exp(-\beta_n)}{1+\exp(-2\beta_n)}(g(x_{2i-2}) + g(x_{2i})))^2)$$

$$+ \sum_{i=1}^{n}(\frac{\lambda^2(1+\exp(-2\beta_n))^2}{2\sigma^4(1-\exp(-2\beta_n))^2}(g(x_{2i-1}) - \frac{\exp(-\beta_n)}{1+\exp(-2\beta_n)}(g(x_{2i-2}) + g(x_{2i})))^2$$

$$\times \sigma^2 \frac{1-\exp(-2\beta_n)}{1+\exp(-2\beta_n)})$$

$$= \exp((\lambda^2 - \lambda)\frac{1+\exp(-2\beta_n)}{2\sigma^2(1-\exp(-2\beta_n))} \sum_{i=1}^{n}(g(x_{2i-1}) - \frac{\exp(-\beta_n)}{1+\exp(-2\beta_n)}(g(x_{2i-2}) + g(x_{2i})))^2).$$

Meanwhile,

$$\mathbb{E}_0[\exp(\lambda \ln \frac{\mathbb{P}_1(f_{10})}{\mathbb{P}_0(f_{10})})] = \mathbb{E}_0[\exp(2\lambda \frac{f(0)g(0) + f(1)g(1) - \exp(-\beta)(f(0)g(1) + f(1)g(0))}{2\sigma^2(1-\exp(-2\beta))}$$

$$- \lambda \frac{g(0)^2 + g(1)^2 - 2\exp(-\beta)g(0)g(1)}{2\sigma^2(1-\exp(-2\beta))})]$$

$$= \mathbb{E}_0[\exp(2\lambda \frac{(g(0) - \exp(-\beta)g(1))f(0) + (g(1) - \exp(-\beta)g(0))f(1)}{2\sigma^2(1-\exp(-2\beta))}$$

$$- \lambda \frac{g(0)^2 + g(1)^2 - 2\exp(-\beta)g(0)g(1)}{2\sigma^2(1-\exp(-2\beta))})]$$

$$= \exp(\frac{4\lambda^2}{8\sigma^4(1-\exp(-2\beta))^2} \text{Var}\,((g(0) - \exp(-\beta)g(1))f(0) + (g(1) - \exp(-\beta)g(0))f(1)))$$

$$- \lambda \frac{g(0)^2 + g(1)^2 - 2\exp(-\beta)g(0)g(1)}{2\sigma^2(1-\exp(-2\beta))})$$

$$= \exp(\frac{4\lambda^2}{8\sigma^4(1-\exp(-2\beta))^2}(\sigma^2(g(0) - \exp(-\beta)g(1))^2 + \sigma^2(g(1) - \exp(-\beta)g(0))^2$$

$$+ 2\exp(-\beta)\sigma^2(g(0) - \exp(-\beta)g(1))(g(1) - \exp(-\beta)g(0)))$$

$$- \lambda \frac{g(0)^2 + g(1)^2 - 2\exp(-\beta)g(0)g(1)}{2\sigma^2(1-\exp(-2\beta))})$$

$$= \exp((\lambda^2 - \lambda)\frac{g(0)^2 + g(1)^2 - 2\exp(-\beta)g(0)g(1)}{2\sigma^2(1-\exp(-2\beta))})).$$

Finally, with ($\heartsuit$) follows by cancelling the $g(i-1)g(i)$ terms, ($\blacklozenge$) by relaxing the exponential terms, and $z(i)$ indicating number of bits before the $i$'s last 1-bit, we have

$$\mathbb{E}_0[\exp(\lambda \ln \frac{\mathbb{P}_1(f_{2^n 0})}{\mathbb{P}_0(f_{2^n 0})})] = \mathbb{E}_0(\exp(\lambda \ln \frac{\mathbb{P}_1(f_{2^{n-1}1}|f_{2^{n-1}0}) \cdot \cdots \cdot \times \mathbb{P}_1(f_{2^0 1}|f_{2^0 0})\mathbb{P}_1(f_{2^0 0})}{\mathbb{P}_0(f_{2^{n-1}1}|f_{2^{n-1}0}) \cdot \cdots \cdot \times \mathbb{P}_0(f_{2^0 1}|f_{2^0 0})\mathbb{P}_0(f_{2^0 0})}))$$

$$= \mathbb{E}_0[\exp(\lambda \ln \frac{\mathbb{P}_1(f_{10})}{\mathbb{P}_0(f_{10})})] \prod_{k=0}^{n-1} \mathbb{E}_0[\exp(\lambda \ln \frac{\mathbb{P}_1(f_{2^k 1}|f_{2^k 0})}{\mathbb{P}_0(f_{2^k 1}|f_{2^k 0})})]$$

$$= \exp(\frac{\lambda^2 - \lambda}{2\sigma^2} \sum_{k=0}^{n-1} \sum_{i=1}^{2^k} \frac{1+\exp(-2\beta/2^k)}{1-\exp(-2\beta/2^k)}(g((2i-1)2^{-(k+1)})$$

$$- \frac{\exp(-\beta/2^k)}{1 + \exp(-2\beta/2^k)}(g((2i-2)2^{-(k+1)}) + g((2i)2^{-(k+1)}))^2$$

$$+ \frac{(\lambda^2 - \lambda)}{2\sigma^2}\frac{1}{1 - \exp(-2\beta)}(g(0)^2 + g(1)^2 - 2\exp(-\beta)g(0)g(1)))$$

$$\overset{(\heartsuit)}{=} \exp(\frac{\lambda^2 - \lambda}{2\sigma^2}(\frac{g(0)^2 + g(1)^2}{1 - \exp(-2\beta)} + \sum_{k=0}^{n-1}\sum_{i=1}^{2^k}\frac{1 + \exp(-2\beta/2^k)}{1 - \exp(-2\beta/2^k)}(g((2i-1)2^{-(k+1)})^2$$

$$+ \frac{\exp(-2\beta/2^k)}{(1 + \exp(-2\beta/2^k))^2}(g((2i-2)2^{-(k+1)})^2 + g((2i)2^{-(k+1)})^2))$$

$$- \sum_{i=1}^{2^n}\frac{2\exp(-\beta/2^{n-1})}{1 - \exp(-2\beta/2^{n-1})}g((i-1)2^{-n})g(i2^{-n})))$$

$$= \exp(\frac{\lambda^2 - \lambda}{2\sigma^2}(\sum_{i=0}^{2^n}(\frac{1 + \exp(-2\beta/2^{z(i)})}{1 - \exp(-2\beta/2^{z(i)})} + 2\sum_{k=z(i)+1}^{n-1}\frac{\exp(-2\beta/2^k)}{1 - \exp(-4\beta/2^k)} - \frac{\exp(-\beta/2^{n-1})}{1 - \exp(-2\beta/2^{n-1})})g(i2^{-n})^2$$

$$+ \sum_{i=1}^{2^n}\frac{\exp(-\beta/2^{n-1})}{1 - \exp(-2\beta/2^{n-1})}g(((i-1)2^{-n}) - g(i2^{-n}))^2) + \frac{g(0)^2 + g(1)^2}{1 - \exp(-2\beta)})$$

$$\overset{(\blacklozenge)}{\leq} \exp(\frac{\lambda^2 - \lambda}{2\sigma^2}(\sum_{i=0}^{2^n}(\frac{1 + \exp(-2\beta/2^{z(i)})}{1 - \exp(-2\beta/2^{z(i)})} + \sum_{k=z(i)+1}^{n-1}2^k/2\beta - 2^{n-1}/2\beta)g(i2^{-n})^2$$

$$+ 2^n/2\beta \times (g(((i-1)2^{-n}) - g(i2^{-n}))^2) + \frac{1}{2}(g(0)^2 + g(1)^2))$$

$$\leq \exp(\frac{\lambda^2 - \lambda}{2\sigma^2}(\sum_{i=0}^{2^n}\beta^2 2^{-n}g(i2^{-n})^2 + \frac{1}{2\beta}2^{-n}((g((i-1)2^{-n}) - g(i2^{-n}))/2^{-n})^2 + \frac{1}{2}(g(0)^2 + g(1)^2)).$$

The lemma follows immediately by letting $\lim n \to \infty$. $\square$

We restate Proposition 9 and prove it. Recall that $g$ is the sample path and $\hat{g}$ the linked list to estimate its evaluations.

**Proposition 9.** *The noised value function (during either training or released) in Algorithm 1 can respond to $N_q$ queries in $\mathcal{O}(N_q \ln(N_q))$ time.*

*Proof.* The value function $Q(\cdot)$ is deterministic. It amounts to show that there is an approach to efficiently estimate the Gaussian process sample path $g$.

We consider the $n$-th query, where the previous $n-1$ queries have been computed and stored. Let $x_1, \ldots, x_{n-1}$ be the previous queries and $g(x_1), \ldots, g(x_{n-1})$ the known value on the sample path. When $x_1 \leq, \ldots, \leq x_{n-1}$, $K_{10}K_{00}^{-1}$ has only two non-zero elements in each row, where the two elements are consecutive. This property holds per the computation in Claim 12, even if $x_i \neq i/2n$. In this case, the two elements need not to be one, but other elements must be zero.

Therefore, the mean $\mu_{at} = K_{10}K_{00}^{-1}$ and the variance $d_{at} = K_{11} - K_{10}K_{00}^{-1}K_{10}^T$ can be computed using these two elements in constant time. The exact calculation is shown below. Thus the noised value function can be calculated in $\mathcal{O}(\ln(N_q))$ time, which is the time complexity of inserting $x_n$ into sorted list $x_1 \leq, \ldots, \leq x_{n-1}$. The proposition follows.

They exact value of $\mu_{at}$ and $d_{at}$ can be verified immediately so we omit the steps of the derivation. Denote, in the linked list $\hat{g}$ in the algorithm, $s^+$ as the element $s$ links to and $s^-$ as the element that links to $s$. Treat $s^+ = 1$ and $s^- = 0$ for non-existence. When $\hat{g}_k[b] = (s, z)$, denote $\hat{g}_k(s) = z$.

Using the arguments above, we have

$$\mu_{at} = \frac{(\exp(\beta(s-s^-)) - \exp(-\beta(s-s^-)))\hat{g}(s^-)}{\exp(\beta(s^+-s^-)) - \exp(-\beta(s^+-s^-))} + \frac{(\exp(\beta(s^+-s)) - \exp(-\beta(s^+-s)))\hat{g}(s^+)}{\exp(\beta(s^+-s^-)) - \exp(-\beta(s^+-s^-))}$$

$$d_{at} = -\frac{(\exp(\beta(s-s^-)) - \exp(-\beta(s-s^-)))\exp(\beta(s-s^-))}{\exp(\beta(s^+-s^-)) - \exp(-\beta(s^+-s^-))}$$

$$-\frac{(\exp(\beta(s^+-s)) - \exp(-\beta(s^+-s)))\exp(\beta(s^+-s))}{\exp(\beta(s^+-s^-)) - \exp(-\beta(s^+-s^-))} + 1.$$

(2)

$\square$

**Claim 12.** *The following equations hold.*

$$K_{11} = \begin{bmatrix} 1 & \exp(-2\beta_n) & \cdots & \exp(-(2n-2)\beta_n) \\ \exp(-2\beta_n) & 1 & \cdots & \exp(-(2n-4)\beta_n) \\ \vdots & \vdots & \ddots & \vdots \\ \exp(-(2n-2)\beta_n) & \exp(-(2n-4)\beta_n) & \cdots & 1 \end{bmatrix},$$

$$K_{10} = \begin{bmatrix} \exp(-\beta_n) & \exp(-\beta_n) & \cdots & \exp(-(2n-1)\beta_n) \\ \exp(-3\beta_n) & \exp(-\beta_n) & \cdots & \exp(-(2n-3)\beta_n) \\ \vdots & \vdots & \ddots & \vdots \\ \exp(-(2n-1)\beta_n) & \exp(-(2n-3)\beta_n) & \cdots & \exp(-\beta_n) \end{bmatrix},$$

$$K_{00} = \begin{bmatrix} 1 & \exp(-2\beta_n) & \cdots & \exp(-2n\beta_n) \\ \exp(-2\beta_n) & 1 & \cdots & \exp(-(2n-2)\beta_n) \\ \vdots & \vdots & \ddots & \vdots \\ \exp(-2n\beta_n) & \exp(-(2n-2)\beta_n) & \cdots & 1 \end{bmatrix}.$$

$$K_{00}^{-1} = \frac{1}{1-\exp(-4\beta_n)} \begin{bmatrix} 1 & -\exp(-2\beta_n) & 0 & \cdots & 0 \\ -\exp(-2\beta_n) & 1+\exp(-4\beta_n) & -\exp(-2\beta_n) & \cdots & 0 \\ 0 & -\exp(-2\beta_n) & 1+\exp(-4\beta_n) & \cdots & 0 \\ \vdots & \vdots & \vdots & \ddots & \vdots \\ 0 & 0 & 0 & \cdots & -\exp(-2\beta_n) \\ 0 & 0 & 0 & \cdots & 1 \end{bmatrix},$$

$$K_{10}K_{00}^{-1} = \frac{\exp(-\beta_n)}{1+\exp(-2\beta_n)} \begin{bmatrix} 1 & 1 & 0 & \cdots & 0 & 0 \\ 0 & 1 & 1 & \cdots & 0 & 0 \\ \vdots & \vdots & \vdots & \ddots & \vdots & \vdots \\ 0 & 0 & 0 & \cdots & 1 & 0 \\ 0 & 0 & 0 & \cdots & 1 & 1 \end{bmatrix},$$

$$K_{10}K_{00}^{-1}K_{10}^T = \frac{\exp(-\beta_n)}{1+\exp(-2\beta_n)} \cdot$$

$$\begin{bmatrix} 2z(1) & z(1)+z(3) & \cdots & z(2n-1)+z(2n-3) \\ z(1)+z(3) & 2z(1) & \cdots & z(2n-3)+z(2n-5) \\ \vdots & \vdots & \ddots & \vdots \\ z(2n-3)+z(2n-5) & z(2n-5)+z(2n-7) & \cdots & z(1)+z(3) \\ z(2n-1)+z(2n-3) & z(2n-3)+z(2n-5) & \cdots & 2z(1) \end{bmatrix},$$

*where we write $z(x) = \exp(-x\beta_n)$ for simplicity.*

$$K_{11} - K_{10}K_{00}^{-1}K_{10}^T = \frac{1-\exp(-2\beta_n)}{1+\exp(-2\beta_n)}\mathbb{I}.$$

*Proof.* The claim may not be obvious, but it can be verified immediately. $\square$

# B    Proof of Proposition 4

The following claim improves Proposition 3 of [HRW13], from the constant 2 to constant 1.25. The claim investigates the anisotropic Gaussian noise mechanism, which can be regarded as a discrete version of the Gaussian process mechanism.

**Claim 13.** *Let $f$ be an vector-input vector-output function. Define the sensitivity under the Mahalanobis distance as $\Delta = \max_{x,x'} \|M^{-1/2}(f(x) - f(x'))\|_2$ where $M$ is positive definite symmetric. Then if $0 < \epsilon < 1$ and $\sigma \geq \sqrt{2\ln(1.25/\delta)}\Delta/\epsilon$, $f(x) + \sigma M^{1/2}y$ is $(\epsilon,\delta)$-DP, where $y$ is drawn from $\mathcal{N}(0, \sigma^2 I)$.*

*Proof.* Let $z \in \mathbb{R}^d$ and $c = \ln(\mathbb{P}(f(x) + \sigma M^{1/2}y = z)/\mathbb{P}(f(x') + \sigma M^{1/2}y = z))$,

$$
\begin{aligned}
c(z) &= \ln \frac{\mathbb{P}(f(x) = z)}{\mathbb{P}(f(x') = z)} \\
&= \frac{(z - f(x))^T M^{-1}(z - f(x))}{2\sigma^2} - \frac{(z - f(x'))^T M^{-1}(z - f(x'))}{2\sigma^2} \\
&= \frac{\|f(x) - f(x')\|_M^2 - 2y^T M^{-1/2}(f(x) - f(x'))}{2\sigma^2}.
\end{aligned}
$$

Hence, when $y \sim \mathcal{N}(0, 1)$,

$$
c(z) \sim \mathcal{N}(\|f(x) - f(x')\|_M^2/2\sigma^2, 2\|f(x) - f(x')\|_M^2/2\sigma^2).
$$

The rest of the argument follows the approach in [DR14] page 261, which is described in the setting of one-dimensional random variables and isotropic $M$. We show the argument in our setting for completeness. For $\delta$-approximation privacy we would like to have $\mathbb{P}(c < \epsilon) > 1 - \delta/2$. We consider the following tail bound of the Gaussian distribution: $\forall t$,

$$
\mathbb{P}(c \geq \mathbb{E}[c] + t) \leq \exp(-t^2/2\operatorname{Var}(c))\sqrt{\operatorname{Var}(c)/2t^2\pi},
$$

which indicates that it is sufficient if both $\ln(\sqrt{2/\pi\delta^2}) \leq \ln(t\sigma/\|f(x) - f(x')\|_M) + t^2\sigma^2/2\|f(x) - f(x')\|_M^2$ and $\|f(x) - f(x')\|_M^2/2\sigma^2 + t \leq \epsilon$ are satisfied for some $t$. The conditions are further reduced to $\ln(\sqrt{2/\pi\delta^2}) \leq \ln(t\sigma/\Delta) + t^2\sigma^2/2\Delta^2$ and $t \leq \epsilon - \Delta^2/2\sigma^2$, respectively. We insert $t = \epsilon - \Delta^2/2\sigma^2$ to the first inequality and derive:

$$
\ln\left(\frac{\epsilon\sigma}{\Delta} - \frac{\Delta}{2\sigma}\right) + \left(\frac{\epsilon^2\sigma^2}{2\Delta^2} + \frac{\Delta^2}{8\sigma^2} - \frac{\epsilon}{2}\right) \geq \ln(\sqrt{2/\pi\delta^2}).
$$

With $\epsilon \leq 1$ we have $\frac{\epsilon\sigma}{\Delta} - \frac{\Delta}{2\sigma} \geq 1$ whenever $\sigma\epsilon/\Delta \geq 3/2$. With $\sigma\epsilon/\Delta \geq 3/2$ we have $\frac{\epsilon^2\sigma^2}{2\Delta^2} + \frac{\Delta^2}{8\sigma^2} - \frac{\epsilon}{2} \geq \sigma^2\epsilon^2/2\Delta^2 - 4/9$ per the monotonicity with respect to $\sigma\epsilon/\Delta$. Hence, it is sufficient that both $\sigma\epsilon/\Delta \geq 3/2$ and $\sigma^2\epsilon^2/2\Delta^2 - 4/9$ are satisfied. The choice $\sigma \geq \sqrt{2\ln(1.25/\delta)}\Delta/\epsilon$ the immediately follows, as desired. $\square$

Now Proposition 4 of this paper follows the Claim 13 above, and Proposition 7 and Proposition 8 of [HRW13].

# C    Proofs of Lemma 6 and Corollary 7

Recall that Lemma 6 finds the desired RKHS and kernel and Corollary 7 is an immediate result that neural networks belong to this RKHS. The key observation is that we cannot restrict the value of $f(0)$ and $f(1)$ to be zero (which is common in some analysis of RKHS), as the value function should not be assumed to have zero value at the boundary.

**Lemma 6.** *We consider the one-dimensional function with bounded variable $x \in \mathbb{R}$, which, without loss of generality, can be treated as $x \in [0, 1]$. The Sobolev space $H^1$ with order 1 and the $\ell^2$-norm (also written as $W^{1,2}$ conventionally) is defined as*

$$
H^1 = \{f \in C[0, 1] : \partial f(x) \text{ exists}; \int_0^1 (\partial f(x))^2 dx < \infty\},
$$

where $\partial f(x)$ denotes weak derivatives and $\int(\cdot)dx$ denotes the Lebesgue integration. If $H^1$ is equipped with inner product

$$\langle f, g \rangle = \frac{1}{2}(f(0)g(0) + f(1)g(1)) + \frac{1}{2\beta}\int_0^1 \partial f(x)\partial g(x) + \beta^2 f(x)g(x)dx$$

where $\beta > 0$, it is an RKHS with kernel $K(x,y) = \exp(-\beta|x-y|)$.

Note that a function being differentiable almost everywhere does not imply that the function has a weak derivative. A counterexample is the Cantor function, which is thus excluded from $H^1$. Any function equal to $\partial f(x)$ almost everywhere is considered identical to $\partial f(x)$ in $H^1$.

*Proof.* The Sobolev space defined in the lemma does not constrain the value to be zero on its border $\{0, 1\}$, hence standard arguments do not apply. However the arguments will be similar. It suffices to show that $f(x)^2 \le c\langle f, f \rangle$ for some $c$ and that $H^1$ is complete. The former can be seen by showing that for any nonzero $f$, $\langle f, g \rangle = 0$ if and only if $g = 0$. By the Cauchy-Schwartz inequality,

$$\langle f, f \rangle \ge \frac{1}{2\beta}\int_0^1 (\partial f(x))^2 dx \ge \frac{1}{2\beta}\int_0^x (\partial f(z))^2 dz$$
$$\ge \frac{1}{2\beta}\left(\int_0^x \partial f(z)dt\right)^2 = \frac{1}{2\beta}f(x)^2.$$

We set $c = 2\beta$ as desired. For the completeness of $H^1$, we show that for any sequence $\{f_n\}$ with $\langle f_n - f_{n+1}, f_n - f_{n+1}\rangle$ converging to zero, the limit of the sequence is in $H^1$. In fact, $\langle f_n - f_{n+1}, f_n - f_{n+1}\rangle$ converging to zero indicates that $\int_0^1 (f_n - f_{n+1})dx$ converges to zero, which then indicates that $\{f_n(x)\}$ converges pointwise for any $x$. The first part of the lemma follows. We then verify that $f(y) = \langle f, K_y \rangle$ for any function $f(y) \in H^1$. In fact,

$$\langle f, K_y \rangle = \frac{1}{2}(f(0)K_y(0) + f(1)K_y(1)) + \frac{\beta}{2}\int_0^1 f(x)K_y(x)dx$$
$$+ \frac{1}{2\beta}(f(x)\partial K_y(x)\big|_0^1 - \int_0^1 f(x)\partial^2 K_y(x)dx)$$
$$= \frac{1}{2}(f(0)K_y(0) + f(1)K_y(1)) + \frac{\beta}{2}\int_0^1 f(x)K_y(x)dx + \frac{1}{2\beta}f(x)(-\beta u(x-y)K_y(x))\big|_0^1$$
$$- \frac{1}{2\beta}\int_0^1 f(x)((-\beta u(x-y))^2 K_y(x) - 2\beta\delta(x)K_y(x))dx)$$
$$= -\frac{1}{2\beta}\int_0^1 -f(x)2\beta\delta(x-y)K_y(x)dx = f(y),$$

where $u(x)$ is the sign function and $\delta(x)$ is the impulse function. By the Riesz representation theorem, $K(x,y)$ is the unique kernel of $H^1$ equipped with the inner product $\langle f, g \rangle$ defined above.  □

**Corollary 7.** *Let $\hat{f}_W(x)$ denote the neural network with finite many finite parameters $W$. For $\hat{f}_W(x)$ with finite many layers, if the gradient of the activation function is bounded, then $\hat{f}_W(x) \in H^1$.*

*Proof.* Let $\psi_i(\cdot)$ be the activation function so the $i$-th layer is represented by the function $\hat{f}_i(x) = \psi(w_i x + b_i)$. Let the gradient of $\psi_i(\cdot)$ be bounded by $c$. As per the chain rule we have

$$\int_0^1 (\partial \hat{f}_W(x))^2 dx \le \prod_{i=1}^{N_c}(f_1(\ldots f_{i-1}(x))\partial f_i|_{x=x_i})^2 < \infty$$

for some $x_i$. Hence $\hat{f}_W(x) \in H^1$.  □

# D Proof of Proposition 10

As the proposition is under the finite state space setting, the context in this section will be different from the rest of the paper. We first restate the proposition and define the notations and preliminaries needed in the analysis. We then show some intermediate claims before we give the final proof.

In the discrete state space setting, we have $\mathcal{S} = \{1, \ldots, n\}$. The stochastic transition kernel is the probability distribution $\mathbb{P}(s'|s, a)$, denoted as the matrices $P_a \in \mathbb{R}^{n \times n}$ (each row sums up to one), $a = 1, \ldots, m$. We write the reward function as $r_a \in \mathbb{R}^n$, $a = 1, \ldots, m$. In this setting, finding the optimal action-state value function is equivalent to finding the optimal state value function $v(s)$, denoted as a vector $v \in \mathbb{R}^n$. The Bellman equation for the optimal value function is given by

$$v \geq \gamma P_i v + r_i, \tag{3}$$

for each $i = 1, \ldots, m$.

**Proposition 10.** *Let $v'$ and $v^*$ be the value function learned by our algorithm and the optimal value function, respectively. In the case $J = 1$, $|S| = n < \infty$, and $\gamma < 1$, the utility loss of the algorithm satisfies*

$$\mathbb{E}[\frac{1}{n}\|v' - v^*\|_1] \leq \frac{2\sqrt{2}\sigma}{\sqrt{n\pi}(1 - \gamma)}.$$

Our utility analysis is based on the linear program formulation under discrete state spaces [DFVR02, CW16] and the sensitivity of linear programs [HRRU14]. In the discrete setting, $v$ is optimal if and only if the Bellman equation (3) is satisfied. In fact, the 'if' relation is immediate, and the 'only if' relation is shown in [SB18] page 64. By exhausting the action set under the max operator and numbering the actions from 1 to $m$, the Bellman equation is formulated into the below linear program:

$$\begin{aligned}
\underset{v}{\text{minimize}} \quad & \mathbf{e}^T v \\
\text{subject to} \quad & (\mathbb{I} - \gamma P_i)v - r_i \geq 0, \quad i = 1, \ldots, m,
\end{aligned} \tag{4}$$

where $\mathbf{e}$ is the all-one vector and $\mathbf{e}^T v$ is the dummy objective. The dual of the linear program (4) is

$$\begin{aligned}
\underset{\lambda_1, \ldots, \lambda_m}{\text{maximize}} \quad & \sum_i \lambda_i^T r_i \\
\text{subject to} \quad & \sum_i (\mathbb{I} - \gamma P_i^T)\lambda_i = \mathbf{e}, \\
& \lambda_i \geq 0, \quad i = 1, \ldots, m.
\end{aligned}$$

We consider the discrete version of Algorithm 1. The Gaussian process noise degenerates to multivariate Gaussian noise. It is also observed that adding noise to the value function is equivalent to adding noise to the reward function, as they are additive in the update. With $J = 1$, it uses the same sample of noise through the training process.

The convergence of the algorithm is guaranteed. In fact, per [SB18] Section 4.4, the value iteration algorithm will converge to the optimal value function of the noised reward function. Formally, given the transition matrices and the noised reward signal $r_i' = r_i + z_i$ for $i = 1, \ldots, m$ where $z_i \sim \mathcal{N}(0, \sigma^2 \mathbb{I})$, Algorithm 1 is guaranteed to converge to a value function $v'$. We desire to show that

$$\mathbb{E}[\frac{1}{n}\|v' - v^*\|_1] \leq \frac{2\sqrt{2}\sigma}{\sqrt{n\pi}(1 - \gamma)},$$

where $v'$ and $v^*$ are the optimal value function under the reward $r'$ and $r$, respectively. $v'$ and $v^*$ are therefore the solution of the system (4) under the reward signal $r'$ and $r$, respectively.

**Lemma 14** ([DFVR02] and [CW16]). *There exists an optimal dual solution $\lambda_i^*$, $i = 1, \ldots, m$, an optimal deterministic policy $\pi^*(\cdot)$, and the corresponding transition matrix $P^*$, such that*

$$\sum_i \lambda_i^* = (\mathbb{I} - \gamma P^{*T})^{-1}\mathbf{e},$$

*and the $k$-th entry of $\lambda_i^*$ equals to the $k$-th entry of $\sum_i \lambda_i^*$ if $\pi^*(k) = i$, and zero otherwise.*

*Proof.* Similar proofs are presented in [DFVR02] and [CW16]. For the completeness of our paper we prove the claim under our context and notations. Denote the superscript $(k)$ as the $k$-th element for a vector and as the $k$-th row for a matrix. Specify $\xi_i^*$ to be any dual optimal solution and construct the policy $\pi^*(k) = \arg\max_i \xi_i^{*(k)}$. Then let

$$\lambda^* = (\mathbb{I} - \gamma P^{*T})^{-1}\mathbf{e},$$

where $P^*$ is the transition matrix of $\pi^*(\cdot)$. The inversion exists since all the eigenvalues of the Markov matrix $P^*$ are smaller than one. Define $\lambda_i^*$, $i = 1, \ldots, m$, such that $\lambda_i^{*(k)} = \lambda^{*(k)}$ whenever $\pi^*(k) = i$ and zero otherwise. We have for $\lambda_i^*$ that

$$\sum_k \sum_i \lambda_i^{(k)}(\mathbb{I} - \gamma P_i)^{(k)} = \mathbf{e},$$

which is a rewrite of the dual feasibility by summing over $k$. We also have $\lambda_i^{*(k)} = 0$ whenever $\xi_i^{*(k)} = 0$ for any $i$ and $k$, and together with the slackness

$$\xi_i^{*T}((\mathbb{I} - \gamma P_i)v - r_i) = 0,$$

we have $\lambda_i^{*T}((\mathbb{I} - \gamma P_i)v - r_i) = 0$. The optimality of $\lambda_i^*$, $i = 1, \ldots, m$ follows. $\qquad\square$

**Claim 15.** *The $\ell^1$-norm of the dual optima $\|\sum_i \lambda_i^*\|_1$ is exactly $n/(1 - \gamma)$.*

*Proof.* By definition we have $\|\sum_i \lambda_i^*\|_1 = \|\lambda^*\|_1$ and $(\mathbb{I} - \gamma P^{*T})\lambda^* = \mathbf{e}$. Since $P^*$ is a Markov matrix, we have $\|P^{*T}\lambda^*\|_1 = \|\lambda^*\|_1$. Taking $\ell^1$-norm and we have $\|\lambda^*\|_1 - \gamma\|\lambda^*\|_1 = \|\mathbf{e}\|_1$. The claim follows. $\qquad\square$

The following lemma justifies that there exists an algorithm to attain bounded suboptimality, given only the noised reward signal. We regard this property as the robustness of the dual system.

**Lemma 16.** *Let $\lambda_i'$, $i = 1, \ldots, m$, be the optimal solution of the system*

$$
\begin{aligned}
\underset{\lambda_1, \ldots, \lambda_m}{\text{maximize}} \quad & \sum_i \lambda_i^T r_i' \\
\text{subject to} \quad & \sum_i (\mathbb{I} - \gamma P_i^T)\lambda_i = \mathbf{e}, \\
& \sum_i \mathbf{e}^T \lambda_i \le \frac{n}{1 - \gamma}, \\
& \lambda_i \ge 0, \quad i = 1, \ldots, m,
\end{aligned}
\tag{5}
$$

*we have*

$$\mathbb{E}[\sum_i \lambda_i'^T r_i] \ge \sum_i \lambda_i^{*T} r_i - \frac{2\sqrt{2}n\sigma}{\sqrt{\pi}(1 - \gamma)}.$$

*Proof.* With ($\heartsuit$) follows the strong duality and ($\blacklozenge$) follows the non-negativity and the convexity, we have

$$
\begin{aligned}
\mathbb{E}[\sum_i \lambda_i'^T r_i] &= \mathbb{E}[\sum_i \lambda_i'^T (r_i' - z_i)] \\
&\ge \mathbb{E}[\sum_i \lambda_i^{*T} r_i' - \sum_i \lambda_i'^T z_i] \\
&= \mathbb{E}[\sum_i \lambda_i^{*T}(r_i + z_i) - \sum_i \lambda_i'^T z_i] \\
&\overset{(\heartsuit)}{=} \sum_i \lambda_i^{*T} r_i + \mathbb{E}[\sum_i (\lambda_i^* - \lambda_i')^T z_i] \\
&\overset{(\blacklozenge)}{\ge} \sum_i \lambda_i^{*T} r_i - \frac{2}{m(1 - \gamma)}\mathbb{E}[\sum_i \|z_i\|_1] \\
&= \sum_i \lambda_i^{*T} r_i - \frac{2\sqrt{2}n\sigma}{\sqrt{\pi}(1 - \gamma)}. \qquad\square
\end{aligned}
$$

It suffices to discuss the connection between the robustness of the primal and the robustness of the dual, which will help us to give a rigorous bound of the utility loss.

Intuitively, if we replace the maximum over $\lambda_i$ by the fixed $\lambda_i'$ in the below derivation of the slackness equation, the subsequent equations will yield $\mathbb{E}[\sum_i \lambda_i'^T r_i]$ which is desired. We observe that relaxing policy optimization (primal) side of the system at the saddle point results in an infeasible point at the value learning (dual) system. It amounts to show that this infeasible point can be mapped to the set of suboptimal values functions. Let $A$ and $B$ be the optimal value of the primal and the dual, the derivation of the slackness equation can be written as

$$
\begin{aligned}
A &= \min_v \max_{\lambda_1,\ldots,\lambda_m \geq 0} \mathbf{e}^T v - (\lambda_1^T((\mathbb{I} - \gamma P_1)v - r_1) + \cdots + \lambda_m^T((\mathbb{I} - \gamma P_m)v - r_m)) \\
&\geq \max_{\lambda_1,\ldots,\lambda_m \geq 0} \min_v \mathbf{e}^T v - (\lambda_1^T((\mathbb{I} - \gamma P_1)v - r_1) + \cdots + \lambda_m^T((\mathbb{I} - \gamma P_m)v - r_m)) \\
&= \max_{\lambda_1,\ldots,\lambda_m \geq 0} \min_v (\lambda_1^T r_1 + \ldots \lambda_m^T r_m) - (-e^T + \lambda_1^T(\mathbb{I} - \gamma P_1) + \cdots + \lambda_m^T(\mathbb{I} - \gamma P_m))v = B.
\end{aligned}
$$

**Claim 17.** *The stochastic policy* $\pi'(i|k) = \lambda_i'^{(k)} / \sum_{i'} \lambda_{i'}'^{(k)}$ *achieves the value* $v'$ *such that* $\mathbf{e}^T v' = \sum_i \lambda_i'^T r_i$.

*Proof.* With Lemma 14 showing the existence, specify $\lambda'' = (\mathbb{I} - \gamma P''^T)^{-1}\mathbf{e}$ and $\lambda_i''$ to be the optimal solution of (5) where $P''$ is the corresponding transition matrix. The Bellman equation indicates that $((\mathbb{I} - \gamma P_i)v' - r_i)^{(k)} = 0$ whenever $\lambda_i''^{(k)} > 0$. It is equivalent to $(\mathbb{I} - \gamma P'')v' - \tilde{r} = 0$ where $\tilde{r}^{(k)} = r_{\pi(k)}^{(k)}$, $k = 1, \ldots, n$. Hence,

$$
\mathbf{e}^T v' = \mathbf{e}^T(\mathbb{I} - \gamma P'')^{-1}\tilde{r} = \tilde{r}^T(\mathbb{I} - \gamma P'')^{-1}\mathbf{e} = \tilde{r}^T \lambda'' = \sum_i \lambda_i'^T r_i. \qquad \square
$$

Armed with the above results, we prove the proposition of the utility guarantee.

*Proof of Proposition 10.* By Lemma 16, our algorithm finds $\lambda_i'$ by solving (5) which satisfies that $\sum_i \lambda_i^{*T} r_i - \mathbb{E}[\sum_i \lambda_i'^T r_i] \leq \frac{2\sqrt{2}n\sigma}{\sqrt{\pi}(1-\gamma)}$. By Claim 17 we have $\mathbb{E}[\sum_i \lambda_i'^T r_i] = \mathbb{E}[\mathbf{e}^T v']$. The strong duality then suggests $\sum_i \lambda_i^{*T} r_i = \mathbf{e}^T v^*$. As $\mathbb{E}[\|v' - v^*\|_1] = \mathbf{e}^T v^* - \mathbb{E}[\mathbf{e}^T v']$, the proposition follows. $\qquad \square$

# E  Details of the Experiments

## E.1  The Environment

The MDP environment is defined as follows: $\mathcal{S} = [0, 1]$ and the state $s$ denotes the location of the agent. $s_0$ is uniformly distributed on $\mathcal{S}$. $\mathcal{A} = \{0, 1\}$. If the agent chooses action 1, the agent will randomly move towards the right by a random amount sampled uniformly from $[0, 0.25]$. If after the move $s$ is greater than 1, it will be reset to 1. Respectively, if the agent chooses action 0, the agent will randomly move towards the left by a random amount sampled uniformly from $[0, 0.25]$. If after the move $s$ is less than 0, it will be reset to 0. The reward $0.5 - |s - 0.5|$ is given at each step, which encourages the agent to move close to the middle of the state space. Each episode of the MDP terminates at the $50^{\text{th}}$ step. The algorithms are trained on 100 episodes or equivalently 5000 samples. The code is available with this manuscript submission.

## E.2  The Baseline Approaches

Balle, Gomrokchi and Precup [BGP16] consider differentially private policy evaluation, where the value function is learned on a one-step MDP using a linear function approximator. This work protects the reward sequence from being distinguishable, but does not ensure the privacy of newly visited states when the value function is released. Thus we do not consider the work as differentially private under our aim of protecting the reward function. Studies on differentially private contextual bandits by Sajed and Sheffet [SS19] and by Shariff and Sheffet [SS18] are considering the equivalent problem, while we use [BGP16] to represent these works.

We also compare with the algorithm proposed by Venkitasubramaniam [Ven13], via input perturbation. In the work, every reward signal is protected by a Gaussian noise thus making the algorithm differentially private. The privacy guarantee is straightforwardly derived by the composition theory by Kairouz, Oh, and Viswanath [KOV13].

We finally compare our algorithm with the differentially private deep learning by Abadi et al. [ACG+16]. As we use a neural network, we can perturb the gradient estimator in the updates such that all inputs are indistinguishable. We use the derived bound in Theorem 1 of [ACG+16]. The $c_2$ constant in that theorem is assigned by $\sqrt{2}$ by following the proof of the theorem.

### E.3 Parameters of Our Approach

We have demonstrated the algorithms on the target of $\epsilon = 0.9$, $\delta = 1 \cdot 10^{-4}$ for Figure 2(a) and $\epsilon = 0.45$, $\delta = 1 \cdot 10^{-4}$ for Figure fig:empirical-compare(b), respectively. In this section we show how exactly these privacy targets are achieved.

Theorem 5 indicates that our algorithm is $(\epsilon, \delta + J \exp(-(2k - 8.68\sqrt{\beta}\sigma)^2/2))$-DP when

$$\sigma \geq \sqrt{2(T/B)\ln(e + \epsilon/\delta)}C(\alpha, k, L, B)/\epsilon,$$

where $C(\alpha, k, L, B) = ((4\alpha(k+1)/B)^2 + 4\alpha(k+1)/B)L^2$, $\beta = (4\alpha(k+1)/B)^{-1}$. We reset the noise on every iteration, namely, let $J = \lfloor T/B \rfloor$. We rewrite the term $J \exp(-(2k - 8.68\sqrt{\beta}\sigma)^2/2)$ as a tight bound $1 - (1 - \exp(-(2k - 8.68\sqrt{\beta}\sigma)^2/2))^J$, which is the probability that all $J$ sample paths are bounded by $2k$. Now we derive the set of parameters.

Let $\delta_g = 1 - (1 - \exp(-(2k - 8.68\sqrt{\beta}\sigma)^2/2))^J$ and $v = (4\alpha(k+1)/B)$. Then $\beta = 1/v$ and $C \approx vL^2$. Plugging in both the values and $T = 5000$, $B = 64$ we have $2k - 8.68\sqrt{\beta}\sigma \approx 2k - 8.6\sqrt{k+1}$. Similar to [ACG+16] we target $1 \times 10^{-4}$-approximation, where it is sufficient if $\delta \leq 5 \times 10^{-5}$ and $\delta_g \leq 5 \times 10^{-5}$. To satisfy $\delta_g \leq 5 \times 10^{-5}$ we need $2k - 8.6\sqrt{k+1} = \ln(1 - \exp(\ln(1 - \delta_g)/J)) \approx 3.5$, when $J = 78$. Thus $k = 23$ will be sufficient. Plugging this $k$ value back to $v$ we have $v = 6.19 \times 10^{-5}$, when $\alpha = 3 \times 10^{-4}$. Finally we target high-privacy regime $\epsilon = 0.9$ and plug in $L^2 = 16$ and have $\sigma \approx \sqrt{2(T/B)\ln(\epsilon/\delta)}vL^2/\epsilon \approx 0.313$.

Approximations are made in the above arguments, but it is immediate to verify that when $\alpha = 3 \times 10^{-4}$, $k = 2.3$, $L^2 = 16$, $B = 64$, $T = 5 \times 10^3$, and $\sigma = 0.32$ the algorithm is $(0.9, 1 \times 10^{-4})$-differentially private. When $\sigma = 0.74$ the algorithm is $(0.45, 1 \times 10^{-4})$-differentially private. The above parameters correspond to Figure 2(a) and 2(b), respectively.