[Reviews · NeurIPS 2019]

Reviewer 1



The authors have significantly improved their paper compared to a previous version. In particular, they have: 1. Made their adjacency definition explicit. 2. Discussed relation to other work. 3. Added experimental comparisons with other methods, even when those are not directly comparable and explained the difference.

Reviewer 2



This paper considers the problem of DP reinforcement learning under continuous state space setting. To be specific, this paper requires DP output of the value function approximator, given neighbouring reward functions. The previous algorithms fail to work since the authors are considering a continuous state space setting. This paper incurs Gaussian process mechanism to add functional noise iteratively in the training and give a theoretical analysis on the privacy guarantees. Utility analysis is also given under discrete state space setting since there exists no theoretical guarantees even for the non-private version of the algorithm under continuous setting. This pape is a nice application of Gaussian process mechanism. Although the algorithm is not complex, yet this paper makes contribution to the community of Differential Privacy, since reinforcement learning is a very important problem in DP. This paper is generally well written although some polishing is preferred. There are some typos and some places are confusing.

Reviewer 3



1. The definition of two neighboring reward functions is provided in Theorem 5. The authors did not explain the motivation of the guarantee of privacy for reward function clearly. It would be better if the authors could interpret the necessities of the privacy of reward function in some real application situations. 2. The description of Algorithm 1 is section 3 is difficult to follow. How about adding noise to the reward function r(.) directly? What is the reason for adding noise like line 19-20 of the Algorithm 1? The definitions of g^_k[B][2] in line 4 and g^_a[:][1] in line 15 are not given. The method to insert s to g^_a[:][1] such that the list remains monotonically increasing in line 15 is also not provided. In line 5 the definition of C(.), k is not given. Proposition 10 is a little confusing. It means with the increasing of the number of iteration rounds, the error also becomes large. The interpretation of proposition 10 should be offered in Section 3. 3. It would be much better if the formal proof of Theorem 5 in page 6 is removed to the appendix, then more words can be used to explain the algorithm thoroughly in the main body. 4. The technique of the paper is based on Gaussian process mechanism.

[Author Response · NeurIPS 2019]

1 We thank all the reviewers for their detailed and positive reviews on our manuscript. We respond to some of the
2 questions and comments below. A further round of polishing has been conducted to improve the quality of the paper.

**1. Motivation and Practical Use Case of the Neighboring Reward Functions**

**Motivation.** As the motivation of our work is to protect the reward function, the mathematical objective is then to make
two reward function $r$ and $r'$ indistinguishable as long as they are 'close' to each other. This 'close' description should
be defined rigorously by some discrepancy measure between functions. The $\ell_\infty$-norm we used is general and natural.
Alternatively, it is also possible to use the distance metric in an RKHS, namely, $\langle r, r' \rangle / \|r\|_{\mathcal{H}} \|r'\|_{\mathcal{H}}$. But this requires
an assumption that $r \in \mathcal{H}$ for some pre-defined $\mathcal{H}$. Hence it is less relevant than the $\ell_\infty$-norm.

**Use case.** The practical use case depends on the exact implementation of the reward function. An example in the
recommendation system: if the system records the clickthrough history of the users and the state $s$ which leads to the
clickthrough, then the reward function can be simulated by using kernel density estimation over $s$ on these clickthroughs.
Then, removing one instance of clickthrough incurs a maximum change of a constant to the infinity norm; Another
example is when the reward function is the average of the utility functions of $N$ users. Removing one user will change
the infinity norm by at most $C_1/N$, as long as these utility functions are bounded by $C_1$.

Overall, our notion of privacy and neighborhood is general enough to be applied to a variety of practical problems.

**2. Explanation of Algorithm 1**

**Adding noise to $r(\cdot)$.** Adding the noise directly to $r(\cdot)$ is the input perturbation method to preserve privacy. Namely, if
we sample $g \in \mathcal{G}(0, \sigma^2 K)$ and replace $r(\cdot)$ in the vanilla deep Q-learning algorithm by $r(\cdot) + g(\cdot)$, then by Proposition
4 the algorithm is differentially private. However, input perturbation is usually less preferred as it tends to incur a high
utility loss. We have illustrated in Figure 2 (blue curve) that it underperforms our algorithm significantly.

**Intuition.** The intuition behind the algorithm is to add functional noise to $Q(\cdot)$. Line 14-18 are an algorithmic
implementation of the Gaussian process (under the Sobolev space and kernel in Lemma 6). More intuitively, we
can regard line 14-18 as generating $g \sim \mathcal{G}(0, \sigma^2 K)$. Then, whenever $Q(s, \cdot)$ is queried (in line 12, 19, and 20),
$Q(s, \cdot) + g(s)$ is returned instead. We have revised our manuscript and commented this intuition on the side of the
algorithm. Therefore the intuition and the discrete implementation will be easier to understand.

**Clarity.** We have made the following revisions for clarity: **A.** In the term $C(\alpha, k, L, B)$ in line 5 of the algorithm, $k$
is a free parameter. It is the tail bound $u/2$ in Lemma 8 that balances the noise level $\sigma$ and the approximation factor
$\delta + J \exp(-(2k - 8.68\sqrt{\beta}\sigma)^2/2)$. For clarity, we have added $k$ to line 2 of Algorithm 1 and then discussed the intuition
$k = u/2$ before Lemma 8. **B.** In line 16 of the algorithm, $\mu_{at}$ and $d_{at}$ are defined in Equation (2), which is in the
appendix. We moved (2) to above Proposition 9 and modified line 16 to *Compute $\mu_{at}$ and $d_{at}$ according to Equation
(2). Then sample $z_{at} \sim \mathcal{N}(\mu_{at}, d_{at})$;.* **C.** $\hat{g}[B][2]$ denotes a linked list of tuple $(s, z)$, pre-allocated with size $B$ of
memory. Whenever a new $s$ is queried, the noise $z$ is calculated in line 16. Then $(s, z)$ is inserted to (already sorted) $\hat{g}$
so that $\hat{g}$ keeps sorted by $s$. Finding the position to insert is done by binary search, namely, *bisect.bisect* in our Python
implementation. **D.** We have shortened the proof of Theorem 5 into a proof sketch to save space for the explanations.

**3. Utility Analysis in Proposition 10**

**Original proposition without $T$.** The number of iteration rounds $T$ is not involved in our Proposition 10. The reason
is that Q-learning algorithms are proved to converge in the discrete state settings. Hence, we consider only the optimal
point that the algorithm will converge to. Denote the optimal points under $r$ and $r'$ as $v^*$ and $v'$, respectively, the utility
analysis investigates how far this perturbed optimal point $v'$ will diverge from the original optimal point $v^*$. Equivalently,
Proposition 10 can be regarded as analyzing the outputs of the algorithm under $r$ and $r'$ by letting $\lim_{T \to \infty}$.

**Proposition with $T$.** Rigorously, we show the utility guarantee with the optimization error. Let $\hat{v}^*$ and $\hat{v}'$ be the actual
output of Algorithm 1 under the true reward $r$ and the neighboring reward $r'$, respectively. By Theorem 1 of Szepesvári's
book [Sze10], under the discrete space $|S| = n < \infty$, $\gamma < 1$, and bounded reward function $\|r(s, a)\|_\infty \leq r_0$, Q-
learning converges in terms of an exponentially decreasing error $2\gamma^{T'} r_0/(1 - \gamma)$ with respect to the number of
iteration rounds $T' = T/B$. By the triangle inequality $\|\hat{v}' - \hat{v}^*\|_1 \leq \|v' - v^*\|_1 + \|\hat{v}' - v'\|_1 + \|\hat{v}^* - v^*\|_1 \leq$
$\|v' - v^*\|_1 + 2n \cdot 2\gamma^{T'} r_0/(1 - \gamma)$. Therefore, Proposition 10 can be re-written as

$$\mathbb{E}[\frac{1}{n}\|\hat{v}' - \hat{v}^*\|_1] \leq \frac{2\sqrt{2}\sigma}{\sqrt{n\pi}(1 - \gamma)} + \frac{4\gamma^{T'} r_0}{1 - \gamma},$$

where the bound is strictly decreasing with the number of iterations rounds $T'$. We believe the confusion by Reviewer
#3 is due to our omitting of the $\mathcal{O}(\gamma^{T'})$ term. As we have revised and added this term back, it should have been clarified.

**References.** [Sze10] Szepesvári, Csaba. "Algorithms for reinforcement learning." Synthesis lectures on artificial
intelligence and machine learning 4, no. 1 (2010): 1-103.

[Meta-Review · NeurIPS 2019]

This paper proposes a differentially private Q-learning algorithm for RL with continuous observations. This is a nice application of the functional Gaussian noise mechanism, and the paper provides a rigorous privacy and utility analysis. When preparing the final version the authors should fix the presentation issues raised in the reviews, and make sure the paper is properly positioned wrt previous work (eg. BGP’16 used a stricter notion of neighbouring relation between datasets that includes changes in the states and actions, not only rewards).